# Pharmacogenomic insights into amlodipine response: the role of *CACNA1D*, *CACNA1C*, and *TRIB3* variants in hypertensive patients

Wahby M. Babaresh[1,2☉*], Zakiullah[1☉*], Sohaib Ahmad Sohail[1], Alija Baig[3‡],
Malik Faisal[4‡], Aiman Begum[1], Haseenullah Shah[1], Zia Ul Hassan[5‡], Kiran Ijaz[6‡],
Syed Muhammad Mukarram Shah[3☉], Aftab Ullah[1,7]

**1** Department of Pharmacy, University of Peshawar, Pakistan, **2** Faculty of Pharmacy, University of Aden, Yemen, **3** Department of Pharmacy, University of Swabi, Swabi, Pakistan, **4** The Lady Reading Hospital Medical Teaching Institution, Peshawar, Pakistan, **5** Cardiology Ward, Hayat Medical Complex Hospital, Peshawar, Pakistan, **6** Institute of Pharmaceutical Sciences, Khyber Medical University, Peshawar, Pakistan, **7** Department of Pharmacy, Abasyn University, Peshawar, Pakistan

☉ These authors contributed equally to this work.
‡ AB, MF, ZUH, and KI authors also contributed equally to this work.
* wahbi.mohammed.pharm@aden-univ.net (WMB), zakiullah@uop.edu.pk (Z)

## Abstract

Hypertension affects over 1.28 billion individuals worldwide, yet response variability to calcium channel blockers (CCBs) like amlodipine remains a challenge. While pharmacogenomic studies have implicated genetic polymorphisms in treatment outcomes, the combined effects of multiple variants remain unclear. This study investigates the influence of *CACNA1D* (rs3774426), *CACNA1C* (rs2239050, rs7311382), and *TRIB3* (rs2295490) variants, individually and in combination, on the antihypertensive response to amlodipine. A total of 133 hypertensive patients from Khyber Pakhtunkhwa, Pakistan, receiving amlodipine monotherapy were genotyped using ARMS-PCR and Sanger sequencing. Blood pressure response was defined as post-treatment systolic blood pressure (SBP) ≤140 mmHg and diastolic blood pressure (DBP) ≤90 mmHg. Statistical analyses were adjusted for age, gender, BMI, dose, family history of HTN and dietary habits. The *CACNA1D* rs3774426 TT genotype was significantly associated with non-response in 35 patients, showing higher SBP than the CC genotype (n=69). Conversely, the *CACNA1C* rs2239050 GG genotype (n=67) was linked to improved SBP and DBP control compared to the CC genotype (n=25). Combined genotype models (*CACNA1D–CACNA1C* and *CACNA1D–CACNA1C–TRIB3*) showed strong unadjusted associations but lost significance after adjustment. These findings highlight the role of *CACNA1D* rs3774426 in predicting amlodipine non-response and demonstrate the potential of genetic screening for optimizing antihypertensive therapy. Integrating pharmacogenomics into clinical practice could enhance personalized treatment strategies, improving outcomes in hypertensive patients.

**Data availability statement:** All relevant data are within the manuscript.

**Funding:** This study was supported by the National Research Programme for Universities (NRPU) under project grant number 17231, titled "Pharmaconomic study of candidate genes involved in selected antihypertensive therapy regimens in patients of Khyber Pakhtunkhwa, Pakistan. The funders had no role in study design, data collection and analysis, decision to publish, or preparation of the manuscript.

**Competing interests:** The authors have declared that no competing interests exist.

## Introduction

Hypertension (HTN) remains a significant public health challenge worldwide due to its high prevalence and associated risks of cardiovascular and renal diseases [1,2]. Epidemiologically, HTN affects approximately one billion individuals globally, and its incidence is increasing in both developed and developing countries [1–3]. Projections suggest that its prevalence will rise by 60% in the next two decades, exceeding 1.5 billion cases by 2030 [1,4]. Despite the availability of effective treatments, only 53% of hypertensive patients achieve adequate blood pressure control due to various factors, including poor adherence, suboptimal dosing, and significant interindividual variability in treatment response which contribute multifactorial disease complexity [5–8]. Similarly, individuals with a family history of HTN are also at increased risk, highlighting the importance of genetic factors in the development of the disease [8,9]. Furthermore, certain medical conditions, such as kidney disease, sleep apnea, and certain endocrine disorders, can also increase the risk of developing HTN [10,11].

Amlodipine, a widely used antihypertensive agent, belongs to the dihydropyridine class of calcium channel blockers (CCBs) [12,13]. It exerts its therapeutic effect by selectively inhibiting L-type calcium channels, primarily located on vascular smooth muscle cells, leading to reduced calcium influx [12,13]. This inhibition causes vasodilation, decreased vascular resistance, and subsequent reduction in blood pressure (BP) [14]. Due to its long half-life and favorable side-effect profile, amlodipine is often prescribed as a first-line treatment for HTN, particularly in patients with comorbid conditions like diabetes or angina [15,16]

The pharmacodynamic effects of amlodipine are significantly impacted by genetic variations in genes encoding components of L-type calcium channels and other related pathways. Genetic polymorphisms in key genes such as *CACNA1D* [17], *CACNA1C*, and *TRIB3* [18] can vary response to antihypertensive therapy of Amlodipine. Among these, the *CACNA1D* rs3774426 polymorphism has been shown to modulate the efficacy of CCBs like amlodipine [17,19]. Patients with the TT genotype exhibit significantly less reduction in systolic blood pressure (SBP) compared to CC or CT genotypes, highlighting their reduced responsiveness to therapy. These findings suggest that rs3774426 may serve as a predictive marker for non-responsiveness in hypertensive patients treated with dihydropyridine CCBs [17,19]

Similarly, polymorphisms in the *CACNA1C* gene, including rs2239050 and rs7311382 [20], play crucial roles in determining treatment outcomes with amlodipine. The rs2239050 GG genotype has been consistently associated with better BP control across various populations compared to the CC or CG genotypes [19–22]. This single nucleotide polymorphism's (SNP) variability in frequency and effect across ethnic groups further underscores the importance of population-specific studies to optimize pharmacogenomic applications. Meanwhile, rs7311382, which exhibits linkage with rs2239050, has been implicated in modulating drug response, particularly among TT genotype carriers who often show diminished efficacy [20]. These polymorphisms likely affect the function of the L-type calcium channels targeted by amlodipine, thereby influencing drug pharmacodynamics [19–22].

Additionally, the *TRIB3* rs2295490 polymorphism has emerged as a significant factor influencing blood pressure response to various anti-hypertensive agents [18]. While AA genotype carriers show greater BP reductions with CCBs like azelnidipine and nitrendipine, AG/GG carriers respond more favorably to angiotensin receptor blockers (ARBs). Mechanistically, *TRIB3* modulates vascular function through its role in the AKT-eNOS-NO signaling pathway; specifically, the G allele of rs2295490 impairs AKT activation, reduces nitric oxide (NO) production, and attenuates vasodilation, potentially contributing to lower efficacy of antihypertensive agents [18]. Stratified analyses reveal that the effects of rs2295490 are further modulated by factors such as age and sex, indicating its potential as a biomarker for personalized therapy [18].

While prior studies have explored the association of *CACNA1D* and *CACNA1C* variants with CCBs response, few have examined these polymorphisms in diverse populations or integrated detailed demographic and lifestyle factors alongside multi-gene interactions. This study addressed these gaps by investigating the impact of genetic, demographic, and lifestyle influences on amlodipine response in a cohort of 133 hypertensive patients from Khyber Pakhtunkhwa, Pakistan. Specifically, we evaluated the individual and combined effects of *CACNA1D* (rs3774426), *CACNA1C* (rs2239050 and rs7311382), and *TRIB3* (rs2295490) polymorphisms on treatment outcomes, uncovering significant genotype-driven differences in systolic and diastolic blood pressure responses. These findings offer novel insights into the pharmacogenomics of CCBs, emphasizing the synergistic potential of *CACNA1D*–*CACNA1C* variants and laying the groundwork for precision medicine in HTN management..

## Methodology

### Study design and population

This cohort study was conducted at the Lady Reading Hospital Medical Teaching Institution (LRH/MTI), Peshawar, enrolling 133 hypertensive patients receiving amlodipine monotherapy between June 2021, and May 2023. Inclusion criteria included individuals aged 30–60 years with a documented diagnosis of primary HTN and adherence to prescribed treatment. Patients with secondary HTN, chronic kidney disease, or concomitant use of other antihypertensive drugs were excluded. The study was approved by the Ethical Review Board of LRH/MTI, Peshawar (Approval Number: 92/LRH/MTI)

### Data collection

Baseline demographic, systolic blood pressure (SBP), and diastolic blood pressure (DBP) data were collected using structured questionnaires and medical records. Written informed consent was obtained from all participants as part of the demographic data collection process, ensuring they were fully informed about the study's objectives, procedures, potential risks, and benefits. Blood pressure measurements were recorded using a calibrated sphygmomanometer, with SBP and DBP values calculated as the mean of three readings taken after a 5-minute rest period..

### Genotyping

Peripheral blood samples were collected from all participants for DNA extraction using the GeneJET Genomic DNA Purification Kit (Thermo Scientific, K0722). The genotyping of the SNPs rs3774426, rs2239050, rs7311382, and rs2295490 was performed using different methodologies. Specifically, rs3774426 was genotyped exclusively by ARMS-PCR, whereas rs2239050, rs7311382, and rs2295490 were genotyped using Sanger sequencing. All allele-specific primers for these SNPs were designed using Primer1 software (from University of Southampton, https://primer1.soton.ac.uk/primer1.html) and NCBI Primer–BLAST, which subsequently validated with tools such as the UCSC Genome Browser, and In-Silico PCR.

For rs3774426, both outer and allele-specific inner primers were used to distinguish between the C and T alleles. For rs2239050 and rs7311382, standard forward and reverse primers were used for amplification of the polymorphic regions within the *CACNA1C* gene. For rs2295490, primers were designed to flank the *TRIB3* variant site. All PCR conditions

were optimized for specificity and efficiency. Detailed information on primer sequences, GC content, melting temperatures (Tm), and amplicon sizes is provided in S1 Table.

The ARMS-PCR reactions for rs3774426 were carried out using DreamTaq Green PCR Master Mix (2X, Thermo Fisher Scientific), which included an initial denaturation at 95°C for 5 minutes, followed by 35 cycles of denaturation at 94°C for 30 seconds, annealing at 60°C for 30 seconds, and extension at 72°C for 1 minute. A final extension step at 72°C for 7 minutes was performed. The amplified products were resolved on 2% agarose gels stained with ethidium bromide and visualized under UV light.

For *CACNA1C* rs2239050 and rs7311382, similar PCR conditions were used: an initial denaturation at 95°C for 5 minutes, followed by 35 cycles of denaturation at 95°C for 30 seconds, annealing at 60°C for 30 seconds, and extension at 72°C for 1 minute, with a final extension at 72°C for 7 minutes. This ensured the robust amplification of the target DNA sequences.

For *TRIB3* rs2295490, genotyping was performed using a protocol described by Zhou et al. (2019). The PCR conditions for this SNP started with an initial denaturation at 94°C for 5 minutes, followed by 36 cycles of denaturation at 94°C for 30 seconds, annealing at 57°C for 30 seconds, and extension at 72°C for 30 seconds, with a final extension at 72°C for 5 minutes. Following PCR, the genetic variants were sequenced using Sanger sequencing at Macrogen Lab in Korea, and according to the manufacturer's instructions.

### Outcome assessment

Treatment response was defined as achieving a post-treatment systolic blood pressure (Post-SBP) ≤140 mmHg and diastolic blood pressure (Post-DBP) ≤90 mmHg, based on the blood pressure targets recommended by the European Society of Cardiology (ESC)/European Society of Hypertension (ESH) blood pressure guidelines [23,24]. Patients were categorized as responders if they met this criterion and as non-responders otherwise. Post-treatment blood pressure was measured using the same standardized protocol as baseline measurements.

### Statistical analysis

Data were analyzed using SPSS version 26.0 (IBM, Armonk, NY, USA). Descriptive statistics were used to summarize baseline characteristics, and comparisons between responders and non-responders were made using chi-square or t-tests as appropriate. Logistic regression was employed to evaluate the association between genetic variants and treatment response, with odds ratios (OR) and 95% confidence intervals (CI) calculated for unadjusted and adjusted models. Adjustments were made for confounders, including age, BMI, gender, dose, family history and dietary habits. Genotype-specific differences in post-treatment SBP and DBP were analyzed using one-way ANOVA.

## Result

### Descriptive study

Table 1 presents demographic and lifestyle characteristics among responders and non-responders to amlodipine. Gender differences were notable, with males comprising 80.8% of responders, while females represented 56.7% of non-responders, suggesting a potential gender influence on treatment response. Age distribution showed the highest percentage of responders in the 30–39 age range (56.2%), while non-responders were evenly split between 30–39 and 40–49 age groups (50.0% each). BMI classification highlighted that 63.3% of non-responders were obese, compared to 27.4% of responders, indicating a possible association between higher BMI and reduced responsiveness. Dietary habits also varied; 70.0% of non-responders had a routine diet, while responders more frequently reported a higher intake of fats and sweets. Financial status differed as well, with 49.3% of responders classified as having good financial standing versus only 20.0% of non-responders, suggesting socioeconomic factors may indirectly impact treatment outcomes.

**Table 1. Demographic and lifestyle characteristics stratified by response status.**

| Variable | Category | Responder N, (%) | Non-responder N, (%) | P value |
|---|---|---|---|---|
| Gender | Male | 59 (80.8) | 26 (43.3) | <0.001 |
| | Female | 14 (19.2) | 34 (56.7) | |
| Age (Year) | 30-39 | 41 (56.2) | 30 (50.0) | 0.103 |
| | 40-49 | 28 (38.4) | 30 (50.0) | |
| | 50-60 | 4 (5.5) | -- | |
| Body Mass Index | Obese | 20 (27.4) | 38 (63.3) | <0.001 |
| | Overweight | 38 (52.1) | 11 (18.3) | |
| | Healthy weight | 15 (20.4) | 11 (18.3) | |
| Marital Status | Married | 63 (86.3) | 46 (76.7) | 0.151 |
| | Unmarried | 10 (13.7) | 14 (23.3) | |
| Obesity | Yes | 45 (61.6) | 46 (76.7) | 0.064 |
| | No | 28 (38.4) | 14 (23.3) | |
| Smoking | Yes | 7 (9.6) | 7 (11.7) | 0.698 |
| | No | 66 (90.4) | 53 (88.3) | |
| Snuffing | Yes | 21 (28.5) | 27 (45.0) | 0.052 |
| | No | 52 (71.2) | 33 (55.3) | |
| Dietary Pattern | Routine diet intake | 40 (54.8) | 42 (70.0) | 0.002 |
| | High fatty food intake | 13 (17.8) | -- | |
| | High fats and sweets intake | 20 (27.4) | 18 (30.0) | |
| Exercise | Yes | 8 (11.0) | 6 (10.0) | 1.000 |
| | No | 65 (89.0) | 54 (90.0) | |
| Financial Status | Good | 36 (49.3) | 12 (20.0) | 0.001 |
| | Average | 26 (35.6) | 21 (35.0) | |
| | Below average | 11 (15.1) | 27 (45.0) | |
| HTN Awareness | Yes | 60 (82.2) | 45 (75.0) | 0.311 |
| | No | 13 (17.8) | 15 (25.0) | |
| Onset of HTN (Year) | 30-39 | 41 (56.2) | 30 (50.0) | 0.373 |
| | 40-49 | 30 (41.1) | 30 (50.0) | |
| | 50-59 | 2 (2.7%) | -- | |
| Duration of HTN (Month) | 0-6 | 19 (26.0) | 7 (11.7) | 0.007 |
| | 7-12 | 40 (54.8) | 31 (51.7) | |
| | 13-18 | 4 (5.5) | 8 (13.3) | |
| | 19-24 | 6 (8.2) | 14 (23.3) | |
| | Above 25 | 4 (5.5) | -- | |
| Family HTN | Yes | 64 (87.7) | 49 (81.7) | 0.335 |
| | No | 9 (12.3) | 11 (18.3) | |
| Amlodipine Dose (Mg) | 5 | 54 (74.0) | 53 (88.3) | 0.038 |
| | 10 | 19 (26.0) | 7 (11.7) | |

Table 2 outlines the genotype and allelic frequencies for four SNPs associated with amlodipine response, comparing these distributions between responders and non-responders. All variants adhere to Hardy-Weinberg Equilibrium (HWE), supporting the stability of observed distributions and validating their applicability for pharmacogenomic analysis.

**Table 2. Genotype and allele frequency distribution in the study population.**

| Gene | SNPs | Genotype\|Allele | Responder N = 73 (%) | Non-responder N = 60 (%) | HWE P-value |
|------|------|------------------|----------------------|---------------------------|-------------|
| CACNA1D | rs3774426 | CC | 49 (67.1) | 20 (33.3) | 0.089 |
| | | CT | 17 (23.3) | 12 (20.0) | |
| | | TT | 7 (9.6) | 28 (46, 7) | |
| | | C allele | 58 (79.5) | 26 (43.3) | 0.777 |
| | | T allele | 15 (20.5) | 34 (56.7) | |
| CACNA1C | rs2239050 | GG | 45 (61.6) | 22 (36.7) | 0.150 |
| | | GC | 21 (28.8) | 20 (33.3) | |
| | | CC | 7 (9.6) | 18 (30.0) | |
| | | G allele | 56 (76.7) | 32 (53.3) | 0.891 |
| | | C allele | 17 (23.3) | 28 (46.7) | |
| | rs7311382 | CC | 47 (64.4) | 28 (46.7) | 0.126 |
| | | CT | 19 (26.0) | 15 (25.0) | |
| | | TT | 7 (9.6) | 17 (28.3) | |
| | | C allele | 57 (78.0) | 36 (60.0) | 0.782 |
| | | T allele | 16 (22.0 | 24 (40.0) | |
| TRIB3 | rs2295490 | AA | 29 (39.7) | 12 (20.0) | 0.637 |
| | | AG | 31 (42.5) | 24 (40.0) | |
| | | GG | 13 (17.8) | 24 (40.0) | |
| | | A allele | 45 (61.6) | 24 (40.0) | 0.811 |
| | | G allele | 28 (38.4) | 36 (60.0) | |

For the *CACNA1D* gene (rs3774426), the CC genotype was more prevalent among responders (67.1%) compared to non-responders (33.3%), suggesting a greater likelihood of favorable response linked to this genotype. Conversely, the TT genotype was observed in a higher percentage of non-responders (46.7%) than responders (9.6%). Allele frequency analysis further highlights that the C allele was predominant among responders (79.5%), whereas the T allele was more common among non-responders (56.7%), underscoring potential allelic associations with drug efficacy. The *CACNA1C* gene (rs2239050) exhibited a notable distribution pattern, with the GG genotype observed in 61.6% of responders compared to only 36.7% of non-responders. This may suggest a link between the GG genotype and improved drug response. In contrast, the CC genotype was more frequent among non-responders (30.0%), while the G allele was predominant in responders (76.7%) and the C allele in non-responders (46.7%). These findings imply that the *CACNA1C* gene may play a role in modulating response to amlodipine treatment. For rs7311382, the CC genotype was observed in 64.4% of responders and in 46.7% of non-responders, suggesting it may be associated with better drug response. Non-responders exhibited a higher frequency of the T allele (40.0%) relative to responders (22.0%), indicating a possible link between the T allele and decreased treatment response. Finally, *TRIB3* (rs2295490) revealed that the AA genotype was more common among responders (39.7%) than non-responders (20.0%), while the G allele appeared more frequently among non-responders (60.0%) compared to responders (38.4%). This distribution suggests that the G allele may be associated with lower responsiveness to amlodipine, while the A allele could favor response.

## Association study

To assess the independent impact of each SNPs on non-response to amlodipine, Table 3 provides logistic regression analysis with both unadjusted and adjusted odds ratios (OR), controlling for gender, age, BMI, and dose, family history and diet. For the *CACNA1D* rs3774426 variant, individuals with the TT genotype showed a significantly higher likelihood

**Table 3. Genetic variants and their association with response to amlodipine treatment using odds ratio.**

| Gene | Genotype /Allele | Responder N (%) | Nonresponde N (%) | Unadjusted OR (95% CI) | p-value | Adjusted OR* (95% CI) | p-value |
|---|---|---|---|---|---|---|---|
| CACNA1D rs3774426 | CC | 49 (67.1) | 20 (33.3) | -- | -- | | |
| | CT | 17 (23.3) | 12 (20.0) | 1.729 (0.701-4.269) | 0.235 | 0.359 (0.087-1.485) | 0.157 |
| | TT | 7 (9.6) | 28 (46, 7) | 9.800 (.701-4.269) | < 0.001 | 8.054 (1.797-36.101 | 0.006 |
| | C allele | 58 (79.5) | 26 (43.3) | -- | -- | | |
| | T allele | 15 (20.5) | 34 (56.7) | 5.056 (2.356-10.851) | <0.001 | 2.103 (0.778-5.683) | 0.143 |
| CACNA1C rs2239050 | GG | 45 (61.6) | 22 (36.7) | -- | -- | | |
| | GC | 21 (28.8) | 20 (33.3) | 1.948 (0.878-4.322) | 0.101 | 0.226 (0.049-1.043) | 0.057 |
| | CC | 7 (9.6) | 18 (30.0) | 5.260 (1.914-14.456) | 0.001 | 1.216 (0.326-4.536) | 0.770 |
| | G allele | 56 (76.7) | 32 (53.3) | -- | -- | | |
| | C allele | 17 (23.3) | 28 (46.7) | 2.882 (1.371-6.058) | 0.005 | 1.246 (0.454-3.419) | 0.670 |
| rs7311382 | CC | 47 (64.4) | 28 (46.7) | -- | -- | | |
| | CT | 19 (26.0) | 15 (25.0) | 1.325 (0.582-3.018) | 0.502 | 0.133 (0.023-0.783) | 0.026 |
| | TT | 7 (9.6) | 17 (28.3) | 4.077 (1.504-11.046) | 0.006 | 0.978 (0.249-3.833) | 0.974 |
| | C allele | 57 (78.0) | 36 (60.0) | -- | -- | | |
| | T allele | 16 (22.0) | 24 (40.0) | 2.375 (1.113-5.067) | 0.025 | 0.754 (0.260-2.187) | 0.604 |
| TRIB3 rs2295490 | AA | 14 (41.2%) | 5 (31.3%) | -- | -- | | |
| | AG | 16 (47.1%) | 8 (50%) | 1.871 (0.793-4.414) | 0.153 | 1.124 (0.357-3.532) | 0.842 |
| | GG | 4 (11.8%) | 3 (18.7%) | 4.462 (1.720-11.571) | 0.002 | 1.334 (0.361-4.931) | 0.666 |
| | A allele | 22 (64.7%) | 9 (56.3%) | -- | -- | | |
| | G allele | 12 (35.3%) | 7 (43.8%) | 0.415 (0.206-0.8350 | 0.014 | 1.207 (0.459-3.171) | 0.703 |

(*) Adjustments were made for confounders, including age, BMI, gender, dose, family history of HTN and dietary habits

of non-response compared to those with the CC genotype, with an unadjusted odds ratio (OR) of 9.8 (p<0.001). This association persisted even after adjusting for confounders, yielding an adjusted OR of 8.05 (p=0.006), suggesting the TT genotype as a strong independent predictor of reduced amlodipine responsiveness.

In *CACNA1C*, the results for both rs2239050 and rs7311382 suggest that variants within this gene may impact amlodipine response, though the strength of association varies. Moving to rs2239050, individuals with the CC genotype were more likely to be non-responders than those with the GG genotype, indicated by an unadjusted OR of 5.26 (p=0.001). However, upon adjustment for gender, age, BMI, dose, family history and diet habits, the association weakened substantially (adjusted OR = 1.22, p=0.770), implying that the initial significance may be confounded by external factors, necessitating further examination of rs2239050's role in drug response. On another hand, rs7311382 in *CACNA1C*, the TT genotype displayed an increased risk of non-response relative to the CC genotype, with an unadjusted OR of 4.08 (p=0.006). After adjusting for confounding factors, the association was no longer significant (adjusted OR = 0.98, p=0.974), suggesting that rs7311382 alone may not be a reliable indicator of treatment outcome in this context. The analysis of *TRIB3* rs2295490 revealed a modest association for the GG genotype with non-response, showing an unadjusted OR of 4.46 (p=0.002). However, this relationship diminished following adjustment, with an adjusted OR of 1.33 (p=0.666), indicating potential confounding effects of gender, age, BMI, dose, family history and diet.

To assess the combined effects of genetic variants on amlodipine response, logistic regression analyses were performed on two genotype combinations: Combo_3 Genes (combining *CACNA1D* rs3774426, *CACNA1C* rs2239050, and *TRIB3* rs2295490) and Combo_2 Genes (combining *CACNA1D* rs3774426 and *CACNA1C* rs2239050). Results are

presented in Table 4 alongside individual variant analyses from Table 3, with both unadjusted and adjusted odds ratios (OR) calculated, the latter adjusted for age, gender, BMI, dose, family history and dietary pattern.

For Combo_3 Genes, patients with all reference genotypes (Wild, CC for rs3774426, GG for rs2239050, AA for rs2295490) were compared to those with any variant/mixed genotypes (all Heterozygous/Variant or mixed). Among 23 patients with all reference genotypes, 20 (87.0%) were responders, compared to 53 (48.2%) of 110 with variant/mixed genotypes. The unadjusted OR for non-response in the variant/mixed group was 7.170 (95% CI: 2.014–25.526, p = 0.002), indicating a strong association. However, after adjusting for confounders (gender, age, BMI, dose, family history, and diet habits), the adjusted OR reduced to 0.392 (95% CI: 0.090–1.706, p = 0.212), indicating the association was no longer statistically significant and suggesting potential confounding..

For Combo_2 Genes, combining only *CACNA1D* rs3774426 and *CACNA1C* rs2239050, 36 (83.7%) of 43 patients with all reference genotypes (Wild, CC and GG, respectively) were responders, compared to 37 (41.1%) of 90 with variant/mixed genotypes. The unadjusted OR for non-response was 7.367 (95% CI: 2.959–18.339, p < 0.001), demonstrating a highly significant association. After adjustment, the OR declined to 2.550 (95% CI: 0.856–7.590, p = 0.093), indicating a trend toward association, although not statistically significant..

Individual variant analysis from Table 3 confirmed that the *CACNA1D* rs3774426 TT genotype remained the strongest predictor of non-response (adjusted OR: 8.054, 95% CI: 1.797–36.101, p = 0.006). In contrast, *CACNA1C* rs2239050 CC (adjusted OR: 1.216, p = 0.770), rs7311382 TT (adjusted OR: 0.978, p = 0.974), and *TRIB3* rs2295490 GG (adjusted OR: 1.334, p = 0.666) showed no significant associations after adjustment. The Combo_3 Genes model lost predictive power upon adjustment (adjusted OR: 0.392, p = 0.212), suggesting no added value over individual variants. Meanwhile, the Combo_2 Genes model showed improved discrimination (adjusted OR: 2.550, p = 0.093), with classification accuracy rising from 66.9% (unadjusted) to 72.9% (adjusted), supporting its potential utility in multi-gene prediction.

One-way ANOVA was conducted to assess genotype-specific differences in post-treatment systolic blood pressure (SBP) and diastolic blood pressure (DBP) across individual (*CACNA1D* rs3774426, *CACNA1C* rs2239050) and combined genotypes (Combo_3 Genes and Combo_2 Genes) in 133 hypertensive patients receiving amlodipine monotherapy (Table 5).

For *CACNA1D* rs3774426, significant differences were observed in post-SBP (p = 0.003), with means of 140.00 ± 10.26 mmHg (CC), 143.10 ± 9.04 mmHg (CT), and 147.33 ± 11.05 mmHg (TT). Post-DBP showed no significant variation (p = 0.167), with means of 89.08 ± 5.24 mmHg (CC), 90.57 ± 4.46 mmHg (CT), and 90.86 ± 5.01 mmHg (TT). For *CACNA1C* rs2239050, both SBP (p < 0.001) and DBP (p = 0.009) differed significantly across genotypes, with SBP means of

Table 4. Combined genetic variants and their association with response to amlodipine treatment using odds ratio.

| Gene Combination | Genotype Combination | Responder N (%) | Non-responder N (%) | Unadjusted OR (95% CI) | p-value | Adjusted* OR (95% CI) | p-value |
|---|---|---|---|---|---|---|---|
| Combo_3 Genes[1] | All Wild (CC, GG, AA) | 20 (87.0) | 3 (13.0) | -- | -- | -- | -- |
| | All heterozygous/Variant or Mixed | 53 (48.2) | 57 (51.8) | 7.170 (2.014–25.526) | 0.002 | 0.392 (0.090-1.706) | 0.212 |
| Combo_2 Genes[2] | All Wild (CC, GG) | 36 (83.7) | 7 (16.3) | -- | -- | -- | -- |
| | All heterozygous/Variant or Mixed | 37 (41.1) | 53 (58.9) | 7.367 (2.959–18.339) | <0.001 | 2.550 (0.856-7.590) | 0.093 |

[1]: is combination effect of (CACNA1D rs3774426, CACNA1C rs2239050, TRIB3 rs2295490),

[2]: is combination effect of (CACNA1D rs3774426, CACNA1C rs2239050).

(*) Adjustments were made for confounders, including age, BMI, gender, dose, family history and dietary habits

**Table 5. Impact of individual and combined genotypes on post-treatment systolic and diastolic blood pressure.**

| Genotype | Post-SBP (mmHg) Mean±SD | p-value | Post-DBP (mmHg) Mean±SD | p-value |
|---|---|---|---|---|
| **rs3774426 (*CACNA1D*)** | | 0.003 | | 0.167 |
| CC | 140.00±10.26 | | 89.08±5.24 | |
| CT | 143.10±9.04 | | 90.57±4.46 | |
| TT | 147.33±11.05 | | 90.86±5.01 | |
| **rs2239050 (*CACNA1C*)** | | <0.001 | | 0.009 |
| GG | 138.26±7.88 | | 88.56±4.97 | |
| GC | 146.34±13.35 | | 91.06±5.50 | |
| CC | 148.13±6.53 | | 91.47±3.48 | |
| **Combo_3 Genes** | | 0.006 | | 0.009 |
| All Wild (CC, GG, AA) (Reference) | 137.10±8.25 | | 87.39±5.86 | |
| All heterozygous/Variant or Mixed | 143.76±10.71 | | 90.39±4.73 | |
| **Combo_2 Genes** | | <0.001 | | 0.002 |
| All Wild (CC, GG) (Reference) | 136.28±7.77 | | 87.91±5.09 | |
| All heterozygous/Variant or Mixed | 145.63±10.48 | | 90.81±4.78 | |

138.26±7.88 mmHg (GG), 146.34±13.35 mmHg (GC), and 148.13±6.53 mmHg (CC), and DBP means of 88.56±4.97 mmHg (GG), 91.06±5.50 mmHg (GC), and 91.47±3.48 mmHg (CC).

The combined genotype Combo_3 Genes (all reference: CC, GG, AA vs. variant/mixed) showed significant differences in both SBP (p=0.006) and DBP (p=0.009). The all-reference group (n=23) had a mean SBP of 137.10±8.25 mmHg and DBP of 87.39±5.86 mmHg, compared to 143.76±10.71 mmHg (SBP) and 90.39±4.73 mmHg (DBP) in the variant/mixed group (n=110). Similarly, Combo_2 Genes (all reference: CC, GG vs. variant/mixed) revealed significant differences in SBP (p<0.001) and DBP (p=0.002). The all-reference group (n=43) exhibited means of 136.28±7.77 mmHg (SBP) and 87.91±5.09 mmHg (DBP), versus 145.63±10.48 mmHg (SBP) and 90.81±4.78 mmHg (DBP) in the variant/mixed group (n=90). Total sample means were 142.61±10.61 mmHg (SBP) and 89.87±5.05 mmHg (DBP).

## Discussion

In the current study, we investigated the impact of genetic variants in *CACNA1D* (rs3774426), *CACNA1C* (rs2239050 and rs7311382), and *TRIB3* (rs2295490) on the response to amlodipine, with a particular focus on systolic and diastolic blood pressure (SBP and DBP) reductions as markers of drug efficacy.

For *CACNA1D* rs3774426, our results demonstrated a significant association between the TT genotype and non-responsiveness to amlodipine, with this genotype found in 46.7% of non-responders but only 9.6% of responders. The unadjusted odds ratio (OR) for this genotype was 9.8 (p<0.001), and the adjusted OR was 8.05 (p=0.006), suggesting a robust association that positions the TT genotype as a potential marker of reduced amlodipine efficacy. This aligns with findings by Kamide et al., who identified *CACNA1D* polymorphisms as associated with the antihypertensive response to dihydropyridine CCBs. Their work underscored the significance of *CACNA1D* variants, particularly in predicting BP reduction among patients treated with CCBs, but did not specifically highlight the pronounced non-responsiveness linked to the TT genotype as observed in our study [17]. These results add to a growing body of evidence suggesting that *CACNA1D* polymorphisms could play a crucial role in individualizing antihypertensive therapy, particularly among patients prescribed amlodipine.

In the *CACNA1C* gene, rs2239050 and rs7311382 emerged as noteworthy markers. The CC genotype of rs2239050 was more frequently found in non-responders (30.0%) than responders (9.6%), with an unadjusted OR of 5.26 (p=0.001). However, after adjusting for confounders such as gender, age, BMI, dose, family history and diet, the association

weakened and was no longer significant (adjusted OR = 1.216, p = 0.770). This result is consistent with findings from Masilela et al., who also reported no significant association between rs2239050 and amlodipine response after adjustment in their South African cohort [22]. Similarly, Bremer et al. found that the GG genotype was associated with improved CCBs treatment outcomes, including amlodipine, while the C allele (CC or CG genotypes) was linked to poorer outcomes [21]. These findings align with the trend observed in our study, where the CC genotype appeared to predict non-responsiveness before adjustment. For *CACNA1C* rs7311382, our study found that the TT genotype was associated with non-responsiveness in unadjusted analysis, with an unadjusted OR of 4.077 (p = 0.006). The TT genotype was found in 28.3% of non-responders compared to 9.6% of responders. However, like rs2239050, this association also lost significance after adjustment (adjusted OR = 0.978, p = 0.974). Jing Linde et al. (2012) reported that rs7311382 is in strong linkage disequilibrium with rs2239050, suggesting that their effects may be interconnected [20]. However, they did not identify a direct association between rs7311382 and amlodipine efficacy. Together, these findings indicate that rs2239050 and rs7311382 may have limited utility as standalone pharmacogenomic markers for amlodipine response. Instead, their combined effects and interactions with other genetic and environmental factors warrant further investigation to clarify their role in personalized HTN treatment.

The *TRIB3* gene, rs2295490, also demonstrated potential as a marker for predicting amlodipine response. In this study, the GG genotype of rs2295490 was more prevalent in non-responders (18.7%) compared to responders (11.8%), with an unadjusted OR of 4.462 (p = 0.002). Although this association did not remain significant after adjustment, it indicates that the GG genotype may be linked to decreased amlodipine responsiveness, warranting further investigation. Zhou et al. provided support for these findings, showing that *TRIB3* rs2295490 polymorphisms were associated with differential responses to various antihypertensive agents, including CCBs [18]. Specifically, they found that AA genotype carriers demonstrated greater BP reductions with CCBs whereas AG and GG carriers responded more effectively to angiotensin receptor blockers (ARBs) (*TRIB3*). Our findings align with these patterns, suggesting that *TRIB3* rs2295490 may play a complex role in influencing BP response depending on the antihypertensive class used. This could signify an avenue for tailored therapies, where genotype-based adjustments might improve treatment outcomes.

The combined analysis of *CACNA1D* rs3774426, *CACNA1C* rs2239050, and *TRIB3* rs2295490 (Combo_3 Genes) revealed a significant unadjusted association between carrying any variant alleles and non-response to amlodipine (OR: 7.170, p = 0.002). However, in the adjusted model, the OR declined substantially to 0.392 (p = 0.212), not only losing statistical significance but also suggesting an inverse trend. This notable change emphasizes the influence of confounders—particularly gender, age, BMI, dose, family history, and diet—in modulating the observed association and may indicate a non-additive or complex interaction between these genetic loci that is sensitive to external variables..

The two-gene model (Combo_2 Genes, *CACNA1D* rs3774426 + *CACNA1C* rs2239050) showed similarly strong unadjusted association (OR: 7.367, p < 0.001), with 83.7% response in reference genotypes (CC, GG) vs. 41.1% in variant/mixed. The adjusted OR was 2.550 (p = 0.093), suggesting a potential combined effect approaching significance. This may reflect a cleaner pharmacogenomic signal by excluding *TRIB3*, which had weaker predictive value after adjustment. The high responder rate in the CC-GG group reinforces the possible synergistic role of these two genes in amlodipine efficacy..

This finding is consistent with prior evidence linking *CACNA1D* and *CACNA1C* polymorphisms to CCBs efficacy (Kamide et al., 2009; Bremer et al., 2006), as these genes encode subunits of the L-type calcium channels targeted by amlodipine. The high responder rate in the CC-GG group underscores a potential synergistic protective effect, warranting further exploration of these two genes as a minimal predictive panel.

Comparing combined and individual effects, *CACNA1D* rs3774426 TT's adjusted OR of 8.054 (p = 0.006) surpasses both combination models, suggesting it may be the dominant driver of non-response in this cohort. However, the unadjusted ORs of the combinations (7.170 and 7.367) exceed those of *CACNA1C* rs2239050 CC (5.260) and *TRIB3* rs2295490 GG (4.462), indicating additive or interactive effects when risk alleles co-occur. The loss of significance

post-adjustment in the combinations, unlike rs3774426 TT, may reflect smaller sample sizes in the reference groups (23 and 43 vs. 133 total) or greater confounding complexity in multi-gene models. Notably, classification accuracy remained higher for Combo_2 (72.9% adjusted), suggesting better predictive utility over Combo_3.

These results extend prior pharmacogenomic studies by demonstrating that while individual variants provide actionable insights, combined genotypes—especially *CACNA1D* and *CACNA1C*—may enhance predictive power in a clinical setting.. In our cohort, 82.7% of patients carried at least one variant allele among CACNA1D, CACNA1C, or TRIB3, indicating that a substantial proportion of the hypertensive population may be genetically predisposed to altered amlodipine response. This high prevalence underscores the potential value of combined genotype screening in risk stratification and treatment optimization, particularly in genetically diverse populations. However, the influence of confounders highlights a limitation: genetic effects alone may not fully predict outcomes without integrating lifestyle and demographic data. Future studies should validate these combinations in larger, diverse populations and explore epistatic interactions using haplotype or machine learning approaches to refine their predictive power for precision HTN management.

To further validate these associations, ANOVA was conducted, and the results highlight significant genotype-dependent variations in amlodipine's antihypertensive efficacy, with *CACNA1D* rs3774426, *CACNA1C* rs2239050, and their combinations (Combo_3 Genes, Combo_2 Genes) influencing post-treatment SBP and DBP (Table 5). For *CACNA1D* rs3774426, the significant SBP difference (p = 0.003) reflects a gradient of increasing pressure from CC (140.00 ± 10.26 mmHg) to TT (147.33 ± 11.05 mmHg), consistent with Kamide et al. (2009), who linked *CACNA1D* variants to reduced CCBs efficacy. The lack of DBP significance (p = 0.167) suggests rs3774426's effect is predominantly systolic, aligning with its strong association with non-response (adjusted OR = 8.05, p = 0.006), possibly due to its role in vascular smooth muscle calcium influx modulation.

*CACNA1C* rs2239050 demonstrated broader influence, with highly significant SBP (p < 0.001) and DBP (p = 0.009) differences. The GG genotype's lower SBP (138.26 ± 7.88 mmHg) and DBP (88.56 ± 4.97 mmHg) compared to CC (148.13 ± 6.53 mmHg SBP, 91.47 ± 3.48 mmHg DBP) support Bremer et al. (2006), who associated GG with better CCBs outcomes. This dual effect on SBP and DBP underscores rs2239050's impact on L-type calcium channel function, though its weaker adjusted logistic regression signal (OR = 1.216, p = 0.770) suggests confounding influences not captured in ANOVA.

The combined genotype Combo_3 Genes showed significant SBP (p = 0.006) and DBP (p = 0.009) differences, with the all-reference group (137.10 ± 8.25 mmHg SBP, 87.39 ± 5.86 mmHg DBP) outperforming the variant/mixed group (143.76 ± 10.71 mmHg SBP, 90.39 ± 4.73 mmHg DBP) by 6.66 mmHg (SBP) and 3.00 mmHg (DBP). This aligns with its unadjusted OR of 7.17 (p = 0.002), indicating additive effects of risk alleles across these genes. Combo_2 Genes exhibited even stronger differences (SBP: p < 0.001; DBP: p = 0.002), with the all-reference group (136.28 ± 7.77 mmHg SBP, 87.91 ± 5.09 mmHg DBP) showing a 9.35 mmHg SBP and 2.90 mmHg DBP advantage over the variant/mixed group (145.63 ± 10.48 mmHg SBP, 90.81 ± 4.78 mmHg DBP). This mirrors its unadjusted OR of 7.37 (p < 0.001) and near-significant adjusted OR (2.84, p = 0.056), suggesting a potent synergistic effect between *CACNA1D* and *CACNA1C*, potentially due to their direct roles in calcium channel pharmacodynamics (Linde et al., 2012).

These ANOVA findings confirm *CACNA1D* rs3774426's systolic specificity, *CACNA1C* rs2239050's broader BP impact, and the enhanced predictive power of combined genotypes, particularly *CACNA1D–CACNA1C*. The consistent SBP advantage (6.66–9.35 mmHg) and DBP benefit (2.90–3.00 mmHg) in reference groups support multi-gene profiling for optimizing amlodipine therapy, with Combo_2 Genes emerging as a key marker of response variability. The inclusion of *TRIB3* in Combo_3 Genes, despite its modest individual effect (Zhou et al., 2019), slightly attenuates the DBP difference compared to the two-gene model, suggesting a complex interplay warranting further investigation..

## Conclusion

This study emphasized the pivotal role of genetic, demographic, and lifestyle factors in determining the antihypertensive response to amlodipine among patients from Khyber Pakhtunkhwa, Pakistan. Among the genetic variants analyzed, the

 

*CACNA1D* rs3774426 TT genotype showed a strong and consistent association with non-response, even after adjusting for confounders. Although *CACNA1C* rs2239050 GG was linked to better BP control in unadjusted analysis, its significance diminished after adjustment. Similarly, combined genotype models involving *CACNA1D*, *CACNA1C*, and *TRIB3* revealed strong crude associations but were not statistically significant in adjusted models, highlighting the influence of non-genetic factors. By integrating genetic, demographic (age, gender, BMI), lifestyle (dietary habits), treatment-related (amlodipine dose), and clinical background (family history of HTN) variables, this study addressed key gaps in prior pharmacogenomic research, offering a more comprehensive understanding of variability in drug response. These findings support the implementation of personalized medicine approaches that combine pharmacogenomic screening—particularly for *CACNA1D* variants—with patient-specific clinical, lifestyle, and treatment-related data to optimize antihypertensive treatment outcomes.

## Supporting information

**S1 Table. Primer sequences used for genotyping of targeted SNP.**
(DOCX)

## Acknowledgments

The authors would like to express their sincere gratitude to the National Research Programme for Universities (NRPU), Higher Education Commission of Pakistan.. We also acknowledge and are grateful to all those who participated in this study

## Author contributions

**Conceptualization:** Wahby Mohammed Babaresh, Zakiullah, Syed Muhammad Mukarram Shah.

**Data curation:** Sohaib Ahmad Sohail, Aiman Begum, Haseenullah Shah, Aftab Ullah.

**Formal analysis:** Sohaib Ahmad Sohail, Aiman Begum, Haseenullah Shah, Aftab Ullah.

**Funding acquisition:** Zakiullah.

**Investigation:** Wahby Mohammed Babaresh, Zakiullah, Syed Muhammad Mukarram Shah.

**Methodology:** Wahby Mohammed Babaresh, Zakiullah, Syed Muhammad Mukarram Shah.

**Project administration:** Wahby Mohammed Babaresh, Zakiullah, Syed Muhammad Mukarram Shah.

**Resources:** Alija Baig, Malik Faisal, Zia Ul Hassan, Kiran Ijaz.

**Software:** Alija Baig, Malik Faisal, Zia Ul Hassan, Kiran Ijaz.

**Supervision:** Zakiullah, Alija Baig, Malik Faisal, Zia Ul Hassan, Kiran Ijaz.

**Validation:** Sohaib Ahmad Sohail, Alija Baig, Malik Faisal, Aiman Begum, Haseenullah Shah, Zia Ul Hassan, Kiran Ijaz, Aftab Ullah.

**Visualization:** Sohaib Ahmad Sohail, Alija Baig, Malik Faisal, Aiman Begum, Haseenullah Shah, Zia Ul Hassan, Kiran Ijaz, Aftab Ullah.

**Writing – original draft:** Wahby Mohammed Babaresh, Zakiullah.

**Writing – review & editing:** Wahby Mohammed Babaresh, Zakiullah, Syed Muhammad Mukarram Shah.

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
