## [Decision Letter · Decision Letter 0]

PONE-D-25-13882Pharmacogenomic Insights into Amlodipine Response: The Role of CACNA1D, CACNA1C, and TRIB3 Variants in Hypertensive PatientsPLOS ONE

Dear Dr. Babaresh,

Thank you for submitting your manuscript to PLOS ONE. After careful consideration, we feel that it has merit but does not fully meet PLOS ONE’s publication criteria as it currently stands. Therefore, we invite you to submit a revised version of the manuscript that addresses the points raised during the review process. Your manuscript has been revised by three different reviewers and found technical issues that are important to consider before resubmitting a revised version of this manuscript. Please submit your revised manuscript by Jun 06 2025 11:59PM. If you will need more time than this to complete your revisions, please reply to this message or contact the journal office at plosone@plos.org . Please include the following items when submitting your revised manuscript:

We look forward to receiving your revised manuscript.

Kind regards,

Agustín Guerrero-Hernandez

Academic Editor

PLOS ONE

Journal Requirements:

“This study was supported by the National Research Programme for Universities (NRPU) under project grant number 17231, titled "Pharmaconomic study of candidate genes involved in selected antihypertensive therapy regimens in patients of Khyber Pakhtunkhwa, Pakistan."”

“The authors would like to express their sincere gratitude to the National Research Programme for Universities (NRPU) for their generous funding and support under the project grant number 17231, titled "Pharmaconomic study of candidate genes involved in selected antihypertensive therapy regimens, in patients of Khyber Pakhtunkhwa, Pakistan." This project would not have been possible without their financial assistance and belief in the significance of this research. We also acknowledge and grateful to all those who participated in this study”

“This study was supported by the National Research Programme for Universities (NRPU) under project grant number 17231, titled "Pharmaconomic study of candidate genes involved in selected antihypertensive therapy regimens in patients of Khyber Pakhtunkhwa, Pakistan."”

Reviewers' comments:

Reviewer's Responses to Questions

**Comments to the Author**

1. Is the manuscript technically sound, and do the data support the conclusions?

Reviewer #1: Yes

Reviewer #2: Yes

Reviewer #3: Yes

2. Has the statistical analysis been performed appropriately and rigorously? 

Reviewer #1: Yes

Reviewer #2: Yes

Reviewer #3: Yes

3. Have the authors made all data underlying the findings in their manuscript fully available?

Reviewer #1: Yes

Reviewer #2: Yes

Reviewer #3: Yes

4. Is the manuscript presented in an intelligible fashion and written in standard English?

Reviewer #1: Yes

Reviewer #2: Yes

Reviewer #3: No

5. Review Comments to the Author

Reviewer #1: The investigation reported in this manuscript Entitled “Pharmacogenomic Insights into Amlodipine Response: The Role of CACNA1D, CACNA1C, and TRIB3 Variants in Hypertensive Patients” is addressing the role of genetic polymorphisms in the treatment outcomes of hypertension with amlodipine, focusing on the impact of genetic, demographic, and lifestyle influences on amlodipine response. For this purpose, they studied the influence of CACNA1D (rs3774426), CACNA1C (rs2239050, rs7311382), and TRIB3 (rs2295490) variants. They genotyped using ARMS-PCR and Sanger sequencing and determined blood pressure

They found that the CACNA1D rs3774426 TT genotype was associated with lack of effect of amlodipine in blood pressure while the CACNA1C rs2239050 GG genotype was associated with improved blood pressure, they suggest that integrating pharmacogenomics into clinical practice could enhance personalized treatment strategies and empathize, based in their results, that genetic effects alone may not fully predict outcomes without integrating lifestyle and demographic data.

Minor comments.

1.- Why are those values of blood pressure defined as normal SBP ≤140 mmHg and Post-DBP ≤90 mmHg?. Please add references in this sentence. In table 5, individual and combined genotypes differences in blood pressure are not dramatically seen.

2.- Is there any reason of percentages added in columns instead of additions per rows?, I think is more intuitive to visualize the effect, in example, of gender:

Responder non-responder

Male: 59 (69%) 26 (31%)

Female. 14 (29%) 34 (71%)

As regards the drafting of the paper in question, I found clarity and completeness in the methods described. The statistical analysis also seemed adequate to me.

Reviewer #2: Reviewer’s Comments

PONE-D-25-13882

This manuscript by Babaresh et al. reports the impact of CACNA1D, CACNA1C, and TRIB3 Variants on the response to amlodipine in 133 hypertensive patients from the Lady Reading Hospital Medical Teaching Institution. The authors performed genotyping of SNPs of rs3774426, rs2239050, rs7311382, and rs2295490. Then, they examined the associations between genetic variants and response to amlodipine. Adjustments were made for cofounders age, BMI, gender, and diet.

The authors

Major comments

1. Introduction. Several key sentences of the study lack references, particularly in the third paragraph. The authors include some references at the end of the section, but it is unknown which reference corresponds to the finding cited. Please include the respective references. For example:

a. Genetic polymorphisms in key genes such as CACNA1D, CACNA1C, and TRIB3 can vary response to antihypertensive therapy of Amlodipine. Among these, the CACNA1D rs3774426 polymorphism has been shown to modulate the efficacy of calcium channel blockers (CCBs) like amlodipine.

b. Similarly, polymorphisms in the CACNA1C gene, including rs2239050 and rs7311382, play crucial roles in determining treatment outcomes with amlodipine. The rs2239050 GG genotype has been consistently associated with better BP control across various populations compared to the CC or CG genotypes. This SNP’s variability in frequency and effect across ethnic groups further underscores the importance of population-specific studies to optimize pharmacogenomic applications. Meanwhile, rs7311382, which exhibits linkage with rs2239050, has been implicated in modulating drug response, particularly among TT genotype carriers who often show diminished efficacy.

2. Introduction. “Mechanistically, TRIB3 impacts vascular function through its role in insulin signaling and nitric oxide (NO) production, pathways critical for BP regulation”. According to Zhou et al. 2019, TRIB3 has a deleterious role in nitric oxide production and may generate a lower response to ACE inhibitors by reducing activation of AKT-eNOS-NO. Please be more specific in this sentence.

3. Methodology. Genotyping. A Table with the primers used may be helpful as a supplement.

4. Results are described with precision.

5. Conclusion. The authors emphasize that the study is novel regarding the association of the polymorphisms and diverse populations, lifestyle factors, and demographic characteristics. Unfortunately, this is not reflected in the conclusions of the study. The following conclusion in not clear “By exploring both individual and combined effects of these genes, alongside demographic and lifestyle factors, our findings highlighted significant variability in therapeutic outcomes, with the CACNA1D TT genotype showing reduced responsiveness and the CACNA1C GG genotype linked to better blood pressure control”. How do these factors influence the variants and the response to amlodipine?

Reviewer #3: The study by Babaresh et al. presents a pharmacogenomic analysis aimed at determining the influence of genetic variations in genes encoding L-type calcium channels—specifically CACNA1D (rs3774426), CACNA1C (rs2239050, rs7311382), and TRIB3 (rs2295490)—on the antihypertensive response to amlodipine. This research addresses a highly relevant topic, considering the global prevalence of hypertension and the widespread use of amlodipine as a first-line treatment. However, upon reviewing the manuscript, several concerns were identified, which are outlined below in the order in which they appear.

1) There are several typographical errors throughout the manuscript that need to be corrected. Additionally, some abbreviations are not defined upon first mention, such as SNPs, while others are defined but not consistently used thereafter, such as hypertension and CCBs. There are also instances of incomplete parentheses both in the main text and within the tables, which should be carefully reviewed and corrected.

2) In the outcome assessment section of the Methods, it is mentioned that patients were categorized as responders if they achieved a post-treatment systolic blood pressure (Post-SBP) ≤140 mmHg and diastolic blood pressure (Post-DBP) ≤90 mmHg. However, the basis for establishing this criterion is not explained, and no references are provided to support the choice of these thresholds.

3) In the association study section of the Results, it is unclear which demographic and lifestyle characteristics were used to calculate the regression with adjusted odds ratios (OR). In some parts, only gender, age, BMI, and diet are mentioned, while in other instances, smoking is also included. Furthermore, although Table 1 lists several variables, the rationale for including only certain factors in the association analysis is not explained. It would be important to consider family history of hypertension, as it may have relevant implications for the study due to the focus on genetic variations in calcium channel genes and their role in antihypertensive response. Additionally, the dose of amlodipine (5 mg vs. 10 mg) appears to have a significant effect on treatment response, as shown in the table 1, and should be considered in the analysis.

4) Finally, although the data support the importance of the combined presence of the studied variants in influencing the response to amlodipine treatment, it is not clear to me, whether there are existing studies reporting the percentage of the global population—and specifically of the study population—that carries these combined polymorphisms, which would be valuable information to better understand the clinical relevance of the findings.

6. PLOS authors have the option to publish the peer review history of their article (what does this mean? ). If published, this will include your full peer review and any attached files.

**Do you want your identity to be public for this peer review?** For information about this choice, including consent withdrawal, please see our Privacy Policy .

Reviewer #1: No

Reviewer #2: **Yes: ** David Centurión

Reviewer #3: No

---

## [Author Response · Author response to Decision Letter 1]

4 Jun 2025

editor comment 1: Please ensure that your manuscript meets PLOS ONE's style requirements, including those for file naming:

Response: Needful Done

Editor comment 2: Please state what role the funders took in the study. If the funders had no role, please state: "The funders had no role in study design, data collection and analysis, decision to publish, or preparation of the manuscript."

If this statement is not correct you must amend it as needed. Please include this amended Role of Funder statement in your cover letter; we will change the online submission form on your behalf.

Response: Needful Done [stated in cover letter]

Editor comment 3: Please remove any funding-related text from the manuscrip

Response: Needful Done

Editor comment 4: Let us know how you would like to update your Funding Statement

Response: No update is required, we agree with the current statement

Reviewer 1 comment 1: • Why are those values of blood pressure defined as normal SBP ≤140 mmHg and Post-DBP ≤90 mmHg? Please add references in this sentence

Response: We have now clarified that the thresholds of SBP ≤140 mmHg and DBP ≤90 mmHg are based on the blood pressure targets recommended by the European Society of Cardiology (ESC)/European Society of Hypertension (ESH) blood pressure guidelines. 2 references were added in the "Outcome Assessment" section

Reviewer 1 comment 1: • In Table 5, individual and combined genotypes differences in blood pressure are not dramatically seen.

Response: • We thank the reviewer for this valuable observation. While the absolute differences in post-treatment blood pressure between genotype groups in Table 5 may appear modest, these differences are statistically significant and clinically meaningful. Prior studies have shown that even small reductions in systolic blood pressure (as little as 5–10 mmHg) are associated with significantly lower cardiovascular risk. • We thank the reviewer for this valuable observation. While the absolute differences in post-treatment blood pressure between genotype groups in Table 5 may appear modest, these differences are statistically significant and clinically meaningful. Prior studies have shown that even small reductions in systolic blood pressure (as little as 5–10 mmHg) are associated with significantly lower cardiovascular risk. Importantly, Table 5 demonstrates both the individual and combined effects of the studied genotypes on blood pressure control. The differences in post-treatment blood pressure were more pronounced among patients carrying combined genotypes (p = 0.002), compared to those with individual genotypes alone (p = 0.167). This suggests that the cumulative genetic burden has a stronger influence on amlodipine response than single variants, supporting the relevance of multi-locus pharmacogenomic profiling. Additionally, this table helps to clarify the comparatively limited role of the TRIB3 rs2295490 variant in influencing treatment response. Specifically, when comparing Combo 2 (CACNA1D + CACNA1C variants) to Combo 3 (CACNA1D + CACNA1C + TRIB3 variants), the incremental impact of including TRIB3 was minimal. This suggests that, within this cohort, the CACNA1D and CACNA1C variants are the primary drivers of genotype-associated differences in blood pressure response to amlodipine, while TRIB3 may have a weaker or context-dependent effect.

Reviewer 1 comment 2: • Is there any reason of percentages added in columns instead of additions per rows? I think is more intuitive to visualize the effect, in example, of gender

Response: We sincerely thank the reviewer for this insightful suggestion. We understand that presenting row-wise percentages (e.g., within gender categories) can improve the interpretability of subgroup effects such as sex-based differences in treatment response.

In the current version of Table 1, we chose to display column-wise percentages, calculated within responder and non-responder groups, to highlight how various demographic and lifestyle factors are distributed within each clinical outcome group. This approach aligns with our primary objective of comparing characteristics between response categories.

Reviewer 2 comment 1: Several key sentences in the Introduction lack immediate references

Response: Thank you for pointing this out. We have revised the Introduction to insert references immediately after each statement that cites prior findings, rather than clustering them at the end of the paragraph.

Reviewer 2 comment 2: “Mechanistically, TRIB3 impacts vascular function…” Please be more specific according to Zhou et al. 2019.

Response: We have revised the sentence for accuracy. The new wording specifies that TRIB3 modulates vascular function through its role in the AKT-eNOS-NO signaling pathway; specifically, the G allele of rs2295490 impairs AKT activation, reduces nitric oxide (NO) production, and attenuates vasodilation, potentially contributing to lower efficacy of antihypertensive agents

Reviewer 2 comment 3: A Table with the primers used may be helpful as a supplement

Response: Thank you for this suggestion. We have created a new S1 Table listing primer sequences, including GC content, melting temperatures (Tm), and amplicon sizes

Reviewer 2 comment 4: The conclusion should better explain how lifestyle and demographic factors influence amlodipine response

Response: We have revised the Conclusion to clearly state that demographic (age, gender, BMI), lifestyle (dietary habits), treatment-related (amlodipine dose), and clinical background (family history of hypertension) variables were associated with reduced amlodipine responsiveness, particularly among patients carrying risk alleles. We emphasize the importance of integrating genetic and clinical/lifestyle factors in personalized hypertension management.

Reviewer 3 comment 1: Typographical errors, undefined abbreviations, inconsistent usage, and incomplete parentheses need correction.

Response: We thank the reviewer for this important feedback. In response, we have carefully and thoroughly proofread the manuscript to ensure the following corrections have been made:

o All typographical errors have been identified and corrected.

o All abbreviations (e.g., SNPs, HTN, CCBs) are now defined upon first mention in the text.

o Abbreviations are used consistently throughout the manuscript.

o All instances of incomplete or mismatched parentheses, both in the main text and tables, have been reviewed and corrected.

Reviewer 3 comment 2: Outcome Assessment: rationale for BP cutoff (≤140/90 mmHg) not explained or referenced.

Response: As explained above, we have added a rationale and references to the European Society of Cardiology (ESC)/European Society of Hypertension (ESH) blood pressure guidelines supporting the use of these thresholds.

Reviewer 3 comment 3: It is unclear which variables were included in adjusted logistic regression. Why were family history and amlodipine dose not included?

Response: We thank the reviewer for this insightful comment and apologize for any lack of clarity in the initial version of the manuscript. There was an oversight in the manuscript presentation, where the covariates adjusted for in our logistic regression analyses were not consistently described across the text and tables. In our original analysis, we did adjust for several relevant covariates, including age, gender, BMI, dietary pattern, family history of hypertension, and amlodipine dose. However, this was not clearly reflected in the final version of Table 3 (association of individual gene variants). We have now corrected this and ensured that the covariates are consistently reported throughout the manuscript. In Table 4 (association of combined genotypes), we recalculated the adjusted logistic regression models to include family history and dose, and we have updated the table and corresponding text accordingly. We appreciate the reviewer’s attention to this important detail, which has helped us improve the clarity and accuracy of our statistical reporting.

Reviewer 3 comment 4: Finally, although the data support the importance of the combined presence of the studied variants in influencing the response to amlodipine treatment, it is not clear to me, whether there are existing studies reporting the percentage of the global population—and specifically of the study population—that carries these combined polymorphisms, which would be valuable information to better understand the clinical relevance of the findings.

Response: We thank the reviewer for this valuable comment. We agree that understanding the population frequency of combined genotypes is essential for assessing the clinical relevance of pharmacogenomic findings. To the best of our knowledge, there are currently no published studies that report the global or regional prevalence of the specific combined CACNA1D–CACNA1C–TRIB3 genotype profiles evaluated in our study. However, data from large-scale population databases such as gnomAD and the 1000 Genomes Project indicate that the individual minor allele frequencies of these variants are relatively common, particularly in Asian populations. In our cohort, we observed that 82.7% of participants carried at least one variant allele within the combined gene model, suggesting that a sizable proportion of hypertensive patients in this regional population may be genetically predisposed to altered amlodipine response. We have added this information to the Discussion to strengthen the interpretation of the study’s clinical relevance.

---

## [Decision Letter · Decision Letter 1]

Pharmacogenomic Insights into Amlodipine Response: The Role of CACNA1D, CACNA1C, and TRIB3 Variants in Hypertensive Patients

PONE-D-25-13882R1

Dear Dr. Babaresh,

We’re pleased to inform you that your manuscript has been judged scientifically suitable for publication and will be formally accepted for publication once it meets all outstanding technical requirements.

Kind regards,

Agustín Guerrero-Hernandez

Academic Editor

PLOS ONE

Additional Editor Comments (optional):

Reviewers' comments:

Reviewer's Responses to Questions

**Comments to the Author**

1. If the authors have adequately addressed your comments raised in a previous round of review and you feel that this manuscript is now acceptable for publication, you may indicate that here to bypass the “Comments to the Author” section, enter your conflict of interest statement in the “Confidential to Editor” section, and submit your "Accept" recommendation.

Reviewer #1: All comments have been addressed

Reviewer #2: All comments have been addressed

Reviewer #3: All comments have been addressed

2. Is the manuscript technically sound, and do the data support the conclusions?

Reviewer #1: Yes

Reviewer #2: Yes

Reviewer #3: Yes

3. Has the statistical analysis been performed appropriately and rigorously? 

Reviewer #1: Yes

Reviewer #2: Yes

Reviewer #3: Yes

4. Have the authors made all data underlying the findings in their manuscript fully available?

Reviewer #1: Yes

Reviewer #2: Yes

Reviewer #3: Yes

5. Is the manuscript presented in an intelligible fashion and written in standard English?

Reviewer #1: Yes

Reviewer #2: Yes

Reviewer #3: Yes

6. Review Comments to the Author

Reviewer #1: I have carefully reviewed the revised manuscript and confirm that all of my previous comments and concerns have been addressed by the authors. I am satisfied with their responses and have no additional suggestions.

Reviewer #2: The authors have addressed all the questions raised by the reviewer. I have no further comments.

Reviewer #3: It is recommended that abbreviations not be used in the abstract; however, if they are used, they should be defined

7. PLOS authors have the option to publish the peer review history of their article (what does this mean? ). If published, this will include your full peer review and any attached files.

**Do you want your identity to be public for this peer review?** For information about this choice, including consent withdrawal, please see our Privacy Policy .

Reviewer #1: No

Reviewer #2: **Yes: ** David Centurión

Reviewer #3: No

---

## [Editor Report · Acceptance letter]

PONE-D-25-13882R1

PLOS ONE

Dear Dr. Babaresh,

I'm pleased to inform you that your manuscript has been deemed suitable for publication in PLOS ONE. Congratulations! Your manuscript is now being handed over to our production team.

Kind regards,

on behalf of

Dr. Agustín Guerrero-Hernandez

Academic Editor

PLOS ONE